# Protocol for a conversation-based analysis study: PREVENT-ED investigates dialogue features that may help predict dementia onset in later life

Sofia de la Fuente Garcia,[1] Craig W Ritchie,[2] Saturnino Luz[1]

**To cite:** de la Fuente Garcia S, Ritchie CW, Luz S. Protocol for a conversation-based analysis study: PREVENT-ED investigates dialogue features that may help predict dementia onset in later life. *BMJ Open* 2019;9:e026254. doi:10.1136/bmjopen-2018-026254

¹Usher Institute of Population Health Sciences and Informatics, University of Edinburgh School of Molecular Genetic and Population Health Sciences, Edinburgh, UK
²Centre for Clinical Brain Sciences, Department of Psychiatry, University of Edinburgh, Edinburgh, UK

**Correspondence to**
Sofia de la Fuente Garcia;
sofia.delafuente@ed.ac.uk

## ABSTRACT

**Introduction** Decreasing the incidence of Alzheimer's disease (AD) is a global public health priority. Early detection of AD is an important requisite for the implementation of prevention strategies towards this goal. While it is plausible that patients at the early stages of AD may exhibit subtle behavioural signs of neurodegeneration, neuropsychological testing seems unable to detect these signs in preclinical AD. Recent studies indicate that spontaneous speech data, which can be collected frequently and naturally, provide good predictors for AD detection in cohorts with a clinical diagnosis. The potential of models based on such data for detecting preclinical AD remains unknown.

**Methods and analysis** The PREVENT-Elicitation of Dialogues (PREVENT-ED) study builds on the PREVENT Dementia project to investigate whether early behavioural signs of AD may be detected through dialogue interaction. Participants recruited through PREVENT, aged 40–59 at baseline, will be included in this study. We will use speech processing and machine learning methods to assess how well speech and visuospatial markers agree with neuropsychological, biomarker, clinical, lifestyle and genetic data from the PREVENT cohort.

**Ethics and dissemination** There are no expected risks or burdens to participants. The procedures are not invasive and do not raise significant ethical issues. We only approach healthy consenting adults and all participants will be informed that this is an exploratory study and therefore has no diagnostic aim. Confidentiality aspects such as data encryption and storage comply with the General Data Protection Regulation and with the requirements from sponsoring bodies and ethical committees. This study has been granted ethical approval by the London-Surrey Research Ethics Committee (REC reference No: 18/LO/0860), and by Caldicott and Information Governance (reference No: CRD18048). PREVENT-ED results will be published in peer-reviewed journals.

## INTRODUCTION

The PREVENT Dementia project is a prospective cohort study that aims to identify early signs of dementia. By developing robust disease models for the preclinical stages of neurodegeneration and relating these to

---

### Strengths and limitations of this study

► First study to gather spontaneous dialogue data from subjects at risk of Alzheimer's disease (AD) for predictive modelling.

► Incorporates elements to analyse spatial navigation abilities, which recent evidence suggests may be useful in detecting preclinical AD.

► Task design balances naturalness and control (it elicits spontaneous dialogues, aiming for external validity, while introducing time and topic constrain, aiming for reasonable intersubject comparisons).

► Task does not assess the same range of spatial navigation abilities as three-dimensional-based tasks.

► Recruitment restrictions inherent to the project might hinder our initial sample size target.

---

risk factors and exophenotypes.[1] The data comprise family history of dementia, comprehensive neuropsychological assessment, genetic risk profiles, neuroimaging (structural, functional and metabolic), biological markers (cerebrospinal fluid [CSF], plasma, urine and salivary) and lifestyle variables. The PREVENT-Elicitation of Dialogues (PREVENT-ED) study examines the predictive potential of information extracted from the participant's speech in spontaneous dialogue and assesses its usefulness for screening in relation to those variables.

Recent studies have investigated the use of speech and language analysis as a source of clinical information for monitoring the progress of neurodegenerative diseases.[2] A recent study by Fraser *et al* included semantic, syntactic, information content and acoustic features in a predictive model which obtained 81% accuracy in distinguishing healthy individuals from people with a diagnosis of Alzheimer's dementia.[3] However, this and other studies in this area[4–6] are limited to individuals with a clinical diagnosis of cognitive impairment and therefore offer little

insight into the early stages of neurodegeneration. Moreover, they were cross-sectional in design and drew on relatively small data sets. In contrast, our study is designed to collect data from at-risk healthy individuals, longitudinally, in tandem with the phenotypically rich, ongoing PREVENT Dementia study. In addition, while previous work on Alzheimer's disease (AD) diagnosis based on speech and language has focused on narrative speech monologues (ie, most cases are descriptions of a scene such as the Boston 'cookie theft' picture description task[7]), we will focus conversational data from dialogues.

Dialogue involves a broader range of psychological processes than monologue. This is because, in order to achieve successful communication, speakers need to find a common ground for understanding, which demands coordination and implies alignment and entailment at different levels.[8] Recent work has employed conversational speech features such as repairs, repetitions and turn-taking patterns as predictors of AD.[4 9–11] Our study will collect dialogue data from the PREVENT Dementia study participants and process them for extraction of acoustic and dialogical features from both voice samples and transcribed recordings, in order to create predictive models.

PREVENT-ED will also assess spatial navigation abilities. These abilities appear to be a sensitive early cognitive marker of AD,[12] and prior research provides evidence for the decline of these abilities in mild cognitive impairment.[13] While still inconclusive, studies of spatial navigation abilities in preclinical stages of AD[14] have prompted increased interest in further investigation of how these skills may be affected in the progression of AD.[15] As the PREVENT neuropsychological battery lacks a spatial navigation task,[16] the purpose of our experimental design is (primarily) to elicit natural dialogues, and to assess spatial navigation abilities. These aims are complementary, as the dialogues will be elicited through a discussion over a map-based task, and therefore, the analysis of dialogue transcripts can be used as an additional means of assessing spatial navigation abilities.

## METHODS AND ANALYSIS
### Objectives
The primary objective of this study is to
1. Examine the predictive potential of information extracted from the participant's speech in spontaneous dialogue as well as its usefulness for screening.
   Additionally, we aim to
2. Identify specific speech and dialogue features that can help predict cognitive decline leading to Alzheimer's dementia.
3. Assess the relationship between such features and certain risk factors found in healthy mid-life participants. Data on these risk factors have been collected by the PREVENT Dementia project and include:
   a. History of parental dementia.

b. Apolipoprotein E (ApoE) status; the presence of ApoE allele ε4 is associated with high risk of dementia.[17]
c. Neuropsychological evaluations with the COGNITO battery.[16] The COGNITO test battery has been developed specifically to look across numerous cognitive domains with tests that are not subject to the ceiling effects of tests designed for use in dementia. It is entirely computer based and has been used in numerous conditions to assess cognition including depression, schizophrenia. The inter-rater reliability of this battery has been stablished and compared with other cognitive measurements[18] and it has now been translated into five languages and Chinese underway.
d. Measures of Aβ42 amyloid in plasma and CSF and increases in tau and p-tau (known markers of cognitive decline and AD).[19]
e. Medial temporal lobe atrophy and white matter lesion volume. The medial temporal lobe is an area of the brain known to be shrunken in people with AD.[20]
4. Assess the possible associations between spatial navigation abilities and the aforementioned risk factors.
5. Assess associations between dialogue features and spatial navigation abilities.

### Participants: sample size and power calculations
Participants are middle-aged healthy volunteers, who were first recruited from the Edinburgh cohort of the PREVENT Dementia study, starting in 2015,[1 16] on the basis of their family history of dementia.

PREVENT-ED is offered to all individuals in the Edinburgh site who have had their baseline assessment and are due to their 2-year follow-up as well as prospective new participants entering the PREVENT Dementia project. Hence, if a participant suits PREVENT's inclusion criteria, it will also be recruited for PREVENT-ED without further criteria checks, as long as they agree to it (for more information on PREVENT's exclusion and inclusion criteria, please refer to[1] and.[16] The participant's risk status with regard to the factors listed above will remain unknown to the PREVENT-ED researchers at the time of the assessment, in order to avoid potential experimentation biases. These data will have been recorded by the main PREVENT project on separate assessment dates and will be disclosed to PREVENT-ED researchers when the project reaches the stage of data analysis.

In terms of the number of participants required, a distinction must be made between the primary and the secondary objectives of the study. The secondary objectives involve hypothesis testing, while the primary objective concerns the creation of machine learning models for prediction. While determination of sample sizes is relatively straightforward for the former, it is less so for the latter. Assessment of the hypothesis that spatial navigation abilities differ in neutral, low and high-risk participants will be done through analysis of covariance, taking the score

in the spatial navigation task as the dependent variable. Allison *et al*[14] report a large effect size (partial $\eta^2 = 0.564$) for a similar wayfinding task comparing three groups (participants with negative biomarkers). Therefore, conservatively assuming that we wish to be able to detect a medium effect size of 0.16, with a power of 80% at a significance level $p < 0.05$, we require a minimum of 63 participants. As regards the machine learning modelling objective, as with most studies involving automatic categorisation it is difficult to estimate precisely the optimal sample sizes and measurable effect sizes. A widely adopted method of sample size estimation for relatively simple machine learning algorithms such as Euclidean distance and Fisher linear discriminant functions places lower bounds at $1.2 \times f$ and $1.4 \times f$ instances (participants), respectively, where $f$ is the number of features of the dataset, for an expected probability of misclassification (PMC) at most 50% greater than an asymptotic PMC of 0.1.[21] In past research, we have employed feature sets containing between 62 features (Geneva Minimalistic Acoustic Parameter Set, [GeMAPS][22]); and as many as 6373 low-level speech features (prior to feature set reduction) for similar classification tasks. A motivation for using the GeMAPS feature set is that it will allow comparability with other studies and future replicability, since this is a standardised feature set that has been used in several computational paralinguistics and psycholinguistics tasks, such as affective computing and mood analysis. As we have done in previous work [6], we aim to train combinations of weak classifiers using a small number of speech features, including turn-taking, pause patterns, speech rate, voice energy and voice quality measures. Therefore, using the GeMAPS feature set in a similar manner implies that a minimum of 75 participants will be necessary for a PMC at most 50% greater than a conservative asymptotic PMC of 0.1 (90% accuracy). However, we aim to collect data from larger numbers of participants, which will allow us to experiment with larger sets of speech features.

## Experimental procedure and design

Edinburgh PREVENT Dementia participants who show an interest in our study will receive an Information Sheet. On attendance, a member of the research team will address any potential queries and take informed consent, prior to the experiment.

Essentially, the procedure for data collection consists of recording conversations. The experimental task designed by our group was inspired by Anderson *et al*. Map Task study, in which 'speakers collaborate verbally to reproduce on one participant's map a route printed on the others.[23] In Anderson's Map Task study, participants were assigned alternatively the role of 'information giver' or 'information follower'. The former was given a map with a route drawn on it and asked to give instructions to the latter on how to follow this route on their map, which was nearly identical to the former's map, except for the route marking. Our study differs from this design in that our participants will always act as information givers

(leader) while the researcher will take the follower role. The rationale for this is to control for the potential influence of the role (giver vs follower) on the strategies that participants employ when performing the task. In addition, holding the role of the participant fixed will ensure a level of consistency across the data helping make our conclusions more robust for this particular experiment. Further differences between the PREVENT-ED and the original Map Task study are that in PREVENT-ED both maps are identical, and that participants are able to see each other but not each other's maps. Therefore, we will be collecting non-verbal as well as verbal cues that occur in dialogue through the use of audio recordings while the participants undertake this map task.

The purpose of the task is to generate dialogue interactions that are as naturalistic as possible, while still having some control over the settings. Although there is a trade-off between control and naturalness, this study design focuses on dialogue interaction mechanisms, rather than dialogue content. While other tasks, such as structured conversations, may elicit more naturally generated content, we would not expect such tasks to generate naturalistic dialogue interaction structures because generally the interviewee would be prompted to speak rather than spontaneously engage in turn-taking, pausing and other interactional behaviours. Thus, the imaginary land to be navigated by the participants is not designed to be a demanding cognitive task, but rather it is designed to generate a cooperative storytelling and to enhance engagement through the completion of a creative journey. This is in line with recent psycholinguistics findings that show how tasks based on maps and games enhance participant engagement and generate spontaneous conversations.[24] The experimental design has two parts which will happen consecutively:

1. **Wayfinding:** intended to generate dialogue through a natural interaction between the researcher (follower) and the participant (leader). They both have a two-dimensional map of the same land, with 15 landmarks, but the participant's map has drawn routes (figure 1), whereas the researcher's does not (figure 2). Therefore, they need to work together in order to reach mutual understanding and complete a journey through the land, where several alternative paths are possible. The task requires going through different landmarks and trading for certain items at each of them. A trained researcher will use scripted prompts either querying the participant's choices or providing feedback to create common ground. This part of the task will be audiovisually recorded. The rational for the use of this task lies on the aim for eliciting dialogue in a way that resembles a natural setting. Giving and taking directions is an activity that belongs to routine life as much as to the experimental set-up. Nevertheless this is still a controlled setting in which the follower is a trained researcher who will ensure both task completion and production of dialogic interactions.

by Audrey Yeo 2018

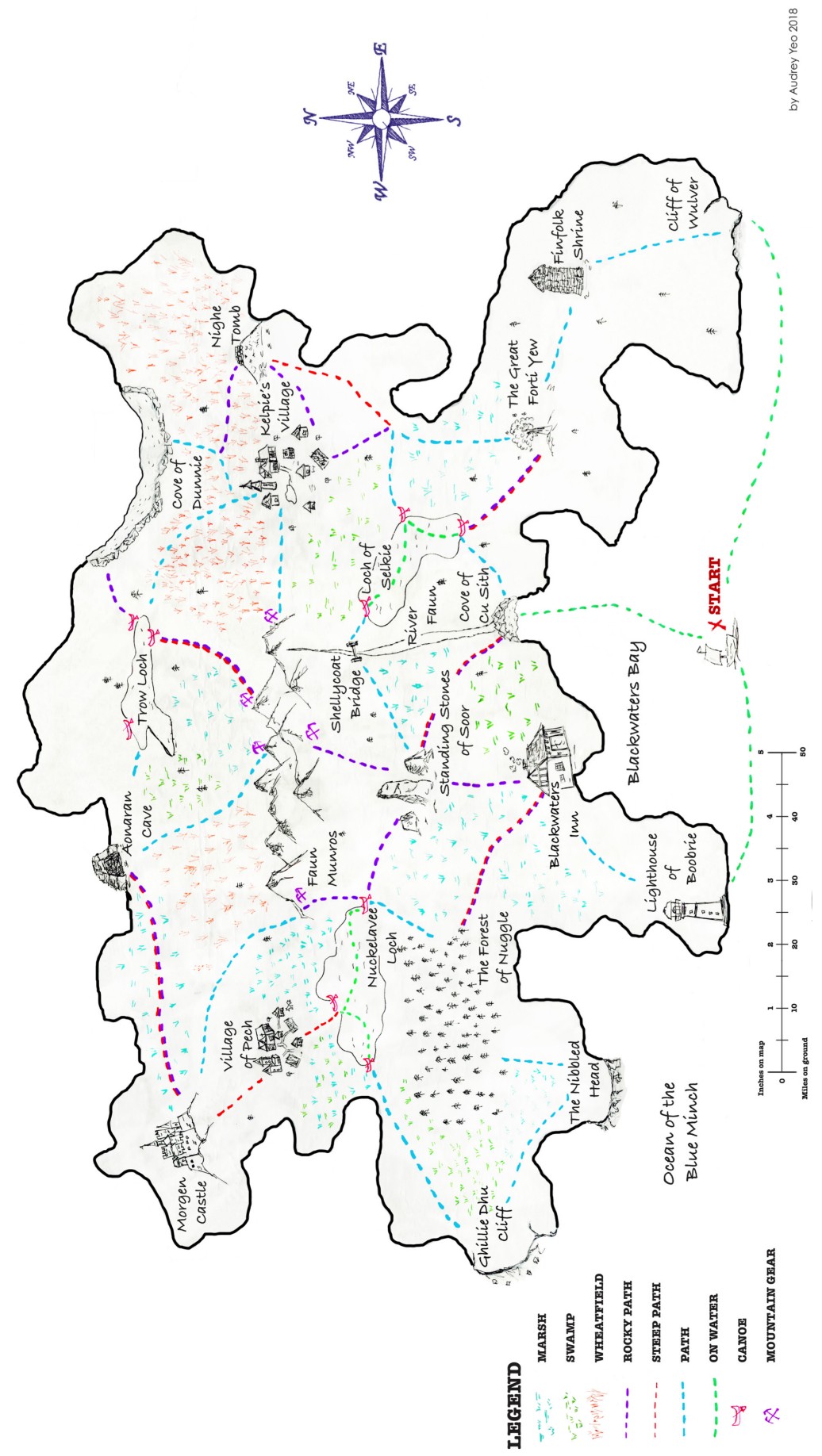

**Figure 1** Wayfinding task: map for the participant (with drawn routes).

by Audrey Yeo 2018

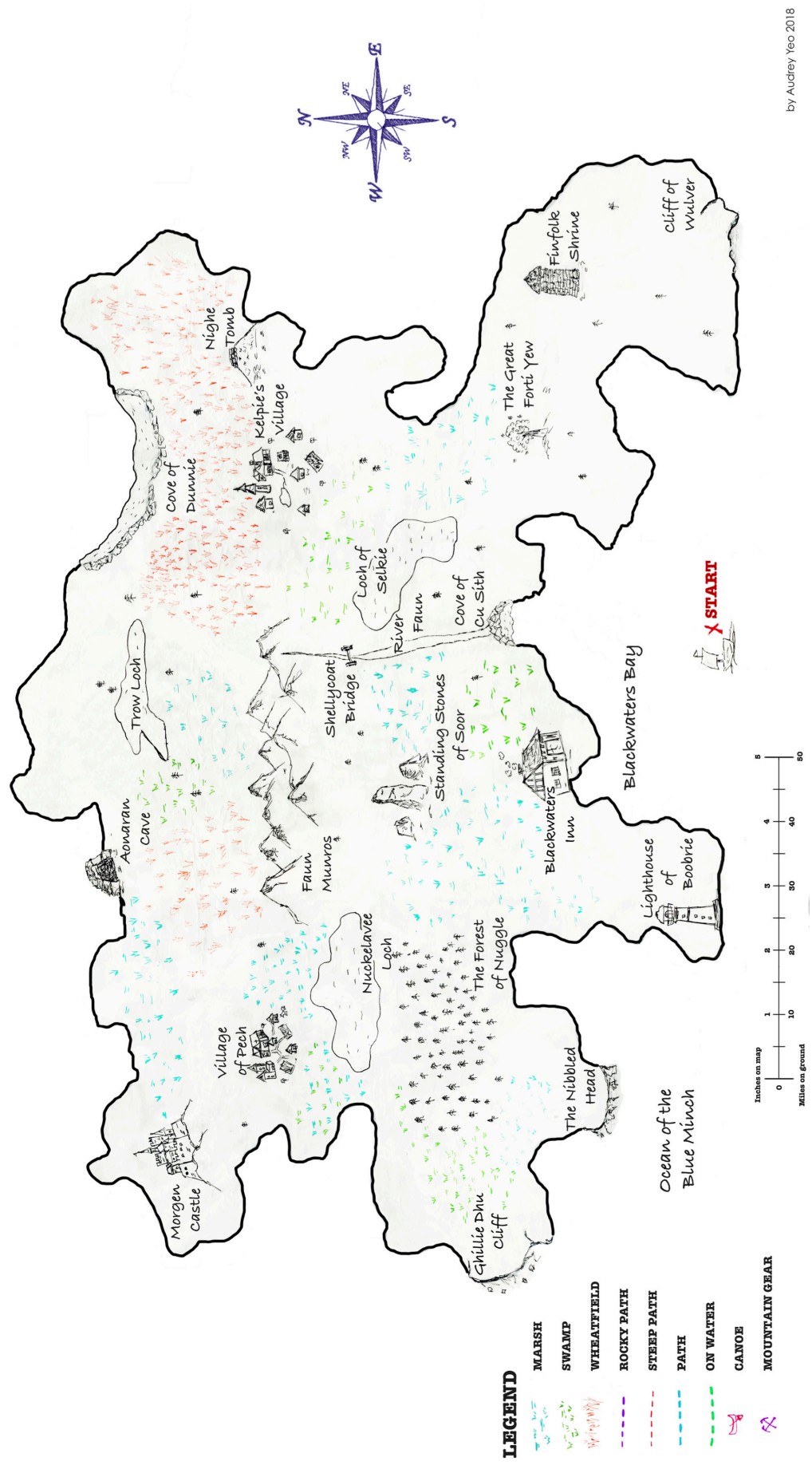

**Figure 2** Wayfinding task: map for the researcher (without drawn routes).

2. **Landmark allocation:** intended to observe the participant's spatial navigation abilities and their spatial memory (ie, memory of spatial information, eg, the layout of the interior of someone's house). After completing the wayfinding task, the participants will receive a version of the map that has neither routes nor landmarks (figure 3) and is asked to place an 'X' at landmark locations on this blank map. Performance is scored in terms of hits and misses on the landmarks they allocate (total: 0–15), as well as the time taken to allocate them. The hit and miss scoring procedure consists on:

a. Scoring 1 point for each correct landmark: following the nearest neighbour criteria, that is, 1 point will be awarded if the participant's choice is closer to the target landmark than to other landmarks, regardless of whether it places on the exact spot.

b. Subtracting 1 point for each missing landmark: if the participant placed less than 15 landmarks.

c. Subtracting 1 point for each misplaced landmark: if a landmark is placed in a different location.

d. Subtracting 0.5 points for 'made-up' landmarks: the participant places more than one landmark where there should be only one, and/or places more than 15 landmarks in total.

This task will always be scored by the same researcher, who will follow these criteria according to research training undertaken prior to data collection. This procedure to assess spatial navigation abilities is based on previous research.[14 25]

During the wayfinding task, the participant does not need to keep in mind the landmarks to be covered on a given route. They only need to focus on giving as much information as possible about the available paths, specifying terrain conditions and discussing distances and choices for directions with the researcher, who is the one signalling which landmark needs to be reached at each stage, and in what order. The potential confounding effect of landmark order during the second task is controlled because it is held fixed by the experimental design of the first task: although there are some voluntary landmarks depending on the chosen alternative path, the journey, narrated by the researcher, goes through the same main landmarks (ie, the trading points), in the same order, for all participants.

The experimental procedure was tested twice, with two different participants, one involved with the main PREVENT project as a research assistant, and another totally unrelated to this research. These preparatory sessions were successful and helped optimise the experimental set-up. They led to the choice of A2 (420×594 mm) as the size of the printed maps, as well as other settings such as lighting, table height, and recorder placement. These sessions also informed logistic decisions such as setting reasonable time slots for each participant to come for the assessment.

## Data management

Conversational data will be recorded by a device developed by our research group specifically for secure collection of speech and video data in healthcare settings, as well as regular close-range microphones. All storage devices are encrypted with state-of-the-art algorithms. Specifically, we employ the Advanced Encryption Standard (AES) with a key size of 256 bits. Dialogues will be transcribed, and ID codes assigned so that only de-identified data will be kept indefinitely within the study database. These procedures are in line with the University of Edinburgh data protection policy, which follows the new General Data Protection Regulation.

There are two main aspects to the data preprocessing: preprocessing for acoustic analysis and preprocessing for natural language processing. For the acoustic analysis, the ELAN (https://tla.mpi.nl/tools/tla-tools/elan/) tool[26 27] will be used for transcription and annotation of dialogue events such as speaker turns, false starts—potentially signalling self-repair events. ELAN is a professional tool developed by the Max Planck Institute for Psycholinguistics, designed for complex annotations on audio and video resources. Audio streams may be studied through different perspectives, from low-level features (eg, frequencies, energy) to medium-level features (eg, syntax, lexicon) or high-level features (eg, sentiment analysis). ELAN allows for hierarchically interconnected annotation tiers which make structured annotations possible in such differentiated levels of analysis. For the natural language processing, ELAN will be used, in transcription mode, with which hierarchical annotations can be synchronised (time-stamped and time-aligned). Annotations are stored in XML format and may also be exported to CHAT, Praat and commonly used text formats for compatibility with other systems and tools.[27]

## Patient and public involvement statement

As these participants conform a subcohort of the PREVENT study cohort, PREVENT-ED benefited from the patient and public involvement measures that were already in place for the main project. The PREVENT Dementia project counts with a participants' panel, established in 2013 prior to the development of the whole research programme. This panel is made up of a group of volunteers who are, themselves, taking part in the study and who meet regularly to discuss the progress of the study and the potential addition of substudies to the project. Two members of this panel also sit in the steering committee. As an additional substudy, PREVENT-ED proposal was presented to this participants' panel, as well as to the prevent steering committee. The potential burdens of the intervention and the time required to participate in the research were assessed and it was decided for the project to be taken forward.

With regard to results disclosure, the procedure within Prevent Dementia is that if something is found which is clinically relevant, it will be fed back to the participant and their general practitioner. This applies to blood

by Audrey Yeo 2018

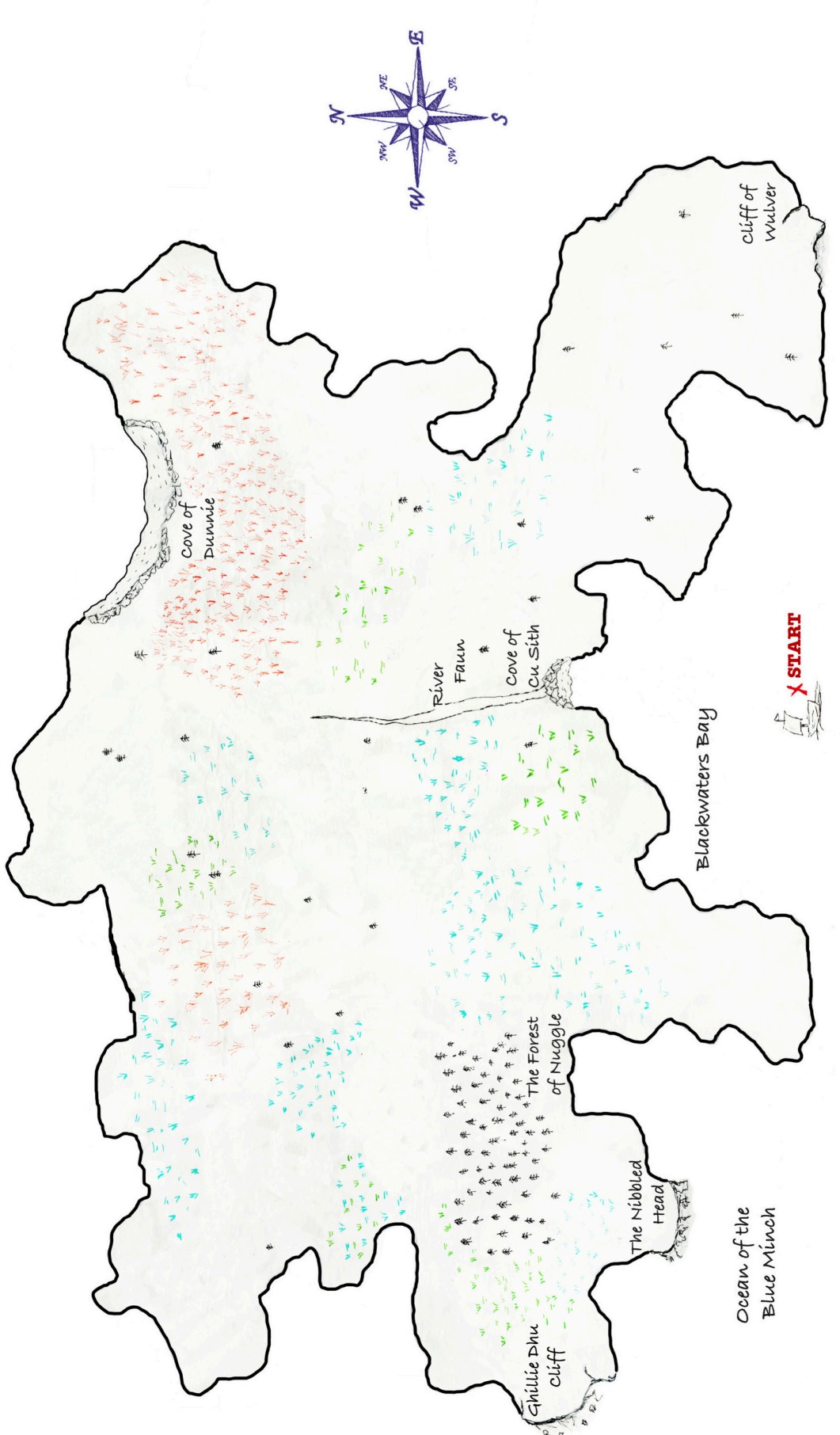

**Figure 3** Landmark allocation task: map for the participant (blank version of the land).

tests, ECG, MRI and validated cognitive tests. However, this does not apply to assessments only carried out for research purposes, as is the case of PREVENT-ED, which is not expected to directly yield clinically relevant results.

## Analysis

The following features will be extracted from the recorded data:

1. GeMAPS for voice research.[22]
2. Alignment of prosodic features (pitch, energy) and speech rate. Alignment in this context means convergence to the same rates for certain speech features. It refers to the psycholinguistic theory that assumes that dialogue processes lead to the automatic coupling of linguistic representations between production and comprehension. This occurs on different levels, and implies accommodation, where the speakers attune to each other throughout the conversation.[8]
3. Dialogue structure features (repair, turn-taking patterns, backchannel behaviour, pauses), to be extracted from ELAN annotations and tiers.[27]
4. Different combinations of:
   a. Voice features: F0, spectral flux, auto-correlation functions (ACF), cepstrum, pitch, onset, beats, energy, voice quality, intensity, vocalisation rhythms.
   b. Content features: mood, sentiment analysis, words, lexical and semantic content.

The spoken dialogue features extracted from the recordings will be regressed over and correlated with data from the neuropsychological evaluations, genetic profiles, biomarkers (amyloid, tau and phosphorylated tau levels in CSF and plaque-dependant inflammation, cortisol levels), neuroimaging (level of brain atrophy in the medial temporal lobe, particularly the hippocampus and entorhinal cortex), family history and spatial navigation abilities.[1 16] In other words, a range of techniques will be applied to investigate whether linguistic, dialogical and paralinguistic features are predictive of, or correlate with:

► Quantitative scores from prevent neuropsychological evaluations (COGNITO battery[16]).
► Categories of ApoE status (presence of ApoE allele ε4 is associated with high risk of dementia).[17]
► Categories of history of parental dementia (0, 1 or 2 parents with a dementia diagnosis).
► Quantitative measures of Aβ42 amyloid in plasma and CSF and increases in tau and p-tau (known markers of cognitive decline and AD).[19]
► Quantitative measures of medial temporal lobe atrophy and white matter lesion volume. The medial temporal lobe is an area of the brain known to be shrunken in people with AD.[20]

Pearson bivariate and multivariate tests will be used to assess simple correlations, and predictive Gaussian process regression will be employed for predictive modelling. This will gauge how much cognitive variance may be explained through these communication patterns, as well

as how much they predict each participant's level of risk or early signs of the disease.

Analytically, we will employ different computational techniques to develop predictions for neurodegenerative decline based on speech features and language. The research team will look for significant differences and use appropriate statistical tests depending on the variables chosen as predictors. Speech signal processing and different machine learning methods, from linear generative classifiers to state-of-the-art deep architectures, will be used to model differences between risk groups. Furthermore, assessment will be ongoing as PREVENT participants are scheduled for a follow-up after at least 2 years and 5 years (with longer term follow-up timespans to be decided). Hence, longitudinal data will eventually be available, including variable outcomes and endpoints where applicable. This will enable us to identify candidate speech markers that could act as early indicators of dementia onset later in life.

## Ethics and dissemination

There are no expected risks or burdens to participants from participating in this study. The procedures do not raise significant ethical issues as they are not invasive and, we only approach healthy consenting adults.

In addition, all participants will be informed that this is an exploratory study and not a diagnostic test. We will assess the extent to which speech 'markers' agree with the score of existing markers and therefore the study cannot find more information than those existing markers. In fact, our research aim is evaluating to what extent this approach would be a good predictor and generate evidence for it.

The main ethical consideration for this study relates to data confidentiality, as it involves collection of audiovisual data, deemed to be identifiable. A discussion with the ethics consultants led us to apply for Caldicott and Information Governance approval and the study complies with the advised requirements regarding data encryption and storage. Also, science and public communications will only include results on analyses undertaken after preprocessing the recordings, ensuring that audiovisual data will never be published or disseminated.

Results from PREVENT-ED will be published in peer-reviewed journals, aiming for an interdisciplinary audience and with a focus on cognitive well-being.

## CONCLUSIONS

PREVENT-ED introduces a novel approach to monitoring early signs of dementia through the analysis of spoken dialogue. While promising results on dialogue analysis have been reported for schizophrenia,[28–30] research on speech in AD has focused more on narrative speech (monologue), both from transcribed recordings[31–33] and from signal processing of voice samples.[34–36] The task introduced in this study aims to elicit dialogue features such as fluency, self-correction, avoidance, pausing behaviour, backchanneling behaviour, question-answering, content

and laughter patterns.[4 37] These features will be extracted from annotations and transcripts, as well as from automatically generated GeMAPS data sets, and will be used for machine learning and statistical analysis to explore their relationship with other risk factors for AD, eventually, their potential to predict preclinical stages of the disease. The map task generates a spontaneous give and take in order to find a common ground for mutual understanding.[23] Even though this interaction is designed to be more spontaneous than a structured interview, the content is still constrained enough so that consistency across data is expected, allowing for comparisons across subjects. In addition, this task will enable us to collect spatial navigation data, which will be investigated along the dialogue features.

Current evidence is scarce regarding which tests are sensitive enough to detect the neurodegeneration that may begin at least 25 years before Alzheimer's dementia is usually diagnosed. The vast majority of studies take place after the onset of Alzheimer's dementia. As we aim to detect signs at earlier stages of neurodegeneration, the PREVENT Dementia dataset offers an ideal platform for our study to identify new relevant associations. Together with our proposed collection of dialogues, the longitudinal analysis of PREVENT Dementia data will add speech-based and conversation-based features to model the preclinical progression of this neurodegenerative disease.

**Acknowledgements** We thank the PREVENT research team for their help and willingness to welcome us in their project, in particular Sarah Gregory, Katie Wells, Clare Dolan and Neil Fullerton. We also acknowledge Audrey Yeo, from the Edinburgh College of Art, for her contribution to the design of the map and storyboard.

**Contributors** SdlFG codesigned the experiment, elaborated the map and the map task's narrative, wrote the initial draft of the paper and revised it following revision and input from coauthors. CWR helped design the experiment, revised and provided feedback on the text and cosupervised the work. SL conceived the idea of using a dialogue task for cognitive state assessment, codesigned the experiment, wrote and revised the text, and cosupervised the work. All authors read and approved the final version of the paper.

**Funding** This work is supported by the Medical Research Council (MRC), grant number MR/N013166/1.

**Competing interests** None declared.

**Patient consent for publication** Not required.

**Ethics approval** This study has been granted approval by the London-Surrey Research Ethics Committee (REC reference No: 18/LO/0860), and by Caldicott and Information Governance (reference No: CRD18048).

**Provenance and peer review** Not commissioned; externally peer reviewed.

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
