## [Reviewer comments · BMJ Open]

ARTICLE DETAILS

TITLE (PROVISIONAL)	Protocol for a conversation-based analysis study: PREVENT-ED investigates dialogue features that may help predict dementia onset in later life
AUTHORS	de la Fuente Garcia, Sofia; Ritchie, Craig; Luz, Saturnino

VERSION 1 - REVIEW

REVIEWER	Frank Rudzicz Associate Professor, University of Toronto, Canada I am an author cited in this work. I am a co-founder of a company that works in this area, called WinterLight Labs.
REVIEW RETURNED	13-Oct-2018

GENERAL COMMENTS	- Some more detail about the “broader range of psychological processes” invoked on page 5 (of 21), line 45, should be provided.- “Interventional research” is invoked (line 19 on page 5 and line 12 on page 7), but there is nothing interventional about this study, nor is the groundwork really laid to point in that direction. Consider removing those claims.- rationale for COGNITO battery above the many others should be given.- Merely “envisioning” (line 18, page 8) 75 participants is insufficient. A proper power analysis should be performed. Rates of atrophy should be identified. It’s worth noting that N=75 is less than the number of participants with AD (never mind the controls) in the “small datasets” (line 36, page 5) to which you refer.- Rationale as to why participants will always be “information givers” (Page 9, line 26) is information that should be given.- More rationale with regards to the utility of way finding above other task elicitation methods should be given- It’s not clear if participants will be meant to attend to landmarks in the first, way finding, task, or in what way the coverage of those landmarks is to be controlled. Surely, the number of landmarks on a route given in the first task should be a covariate to the analysis of the second, at least. The threshold for a ‘hit’ or a ‘miss’ should also be made explicit.- How dialogues are to be transcribed (pg 11, line 11) must be made explicit. You will undoubtedly need to have a strict method to annotate non-speech events, false starts, mispronunciations, cross talk, and speaker turns. E.g., if you intend to use CHAT or some other format or protocol — cite it and explain it briefly.- explain what is meant by ‘alignment’ on page 12, line 9.- explain how dialogue structure features are to be extracted (page 12, line 11)- provide more detail as to the planned regressions and correlations (page 12, line 25)
--

	 - include an appropriate article (the part-of-speech) between 'predict' and 'participant's' on line 31 of page 12 - With regards to Ethics, typically there's a clause where you have to inform participants if an atypical score was computed on the validated (or semi-validated) test (in this case COGNITO). Is this to be done? - whether the constrained content will truly "allow reasonable comparisons across subjects" has not really been demonstrated; this claim may only be evident after data are collected. - the complete inclusion/exclusion criteria must be given. - the maps shown in figures 1-3 have text that may be hard to read, and accoutrement that may serve as visual 'noise' to participants. Either demonstrate a test of visual acuity in your inclusion/exclusion criteria, or make the maps much more minimalist, with clear typeface where appropriate. - One of the prime purposes of this work is to attempt to identify earlier signs of dementia than currently reported in the literature. Alternatively, the question is whether conversation, navigation tasks, or their combination may be useful for that purpose. While laudable, none of the analysis outlined on page 12 will answer that question without very long-term follow-up that has not been explained in this proposal. - It's not clear if preliminary pilot work has been conducted in this setting, with a few experimental participants. That would greatly strengthen this proposal. - BMJ Open expects that "dates of the study should be included in the manuscript." Therefore, the dates of the study should be included in the manuscript.
--	--

REVIEWER	Adam Vogel The University of Melbourne, Australia
REVIEW RETURNED	21-Nov-2018

GENERAL COMMENTS	Please provide more detail on the methods used - additional minor comments attached. what broad acoustic measures are of interest Speech and language are different domains - please acknowledge the difference. speech=motor output, language=content. More detail on speech tasks are warranted. The reviewer provided a marked copy with additional comments. Please contact the publisher for full details.
---

VERSION 1 – AUTHOR RESPONSE

REVIEWER 1

Introduction

- Added to "broader range of psychological processes": coordination, common ground, alignment, entailment at different levels.

- Deleted: claim on "interventional research " claims

Methods and analysis: participants: sample size and power calculation - Added:

- o explanation of sample size, power analysis and size effect
- o with respect to atrophy, all factors listed in the objectives are recorded by the

Prevent main team and will not be disclosed to us until data analysis stage

- o participants are contacted through Prevent main project, hence the exclusion/inclusion criteria for our study is theirs. The only further question they are asked in order to participate in our study is whether they want or not. But if a participant suits PREVENT's inclusion criteria, it suits ours by default.

Methods and analysis: experimental procedure and design - Added:

- o Rationale for wayfinding: eliciting dialogue in a natural setting (we all have given or discussed directions at some point)

- o Rationale for information givers: consistency (limited number of participants) and controlling for confounding effects on task strategy

- o Tasks 1 / 2: task 1 does not require the person to focus on landmarks, since researcher is the one narrating the journey. They need to focus on paths and ways to move through the land. All participants complete the same journey, hence landmarks visited is not a confounder for task 1 (it's controlled by being fixed)

- o Scoring task 2 - hit/miss (from 0 to 15): each correct landmark adds 1 point. Each misplaced or missing landmark deducts 1 point. Each "made-up" landmark deducts 0.5 points. It will always be the same researcher scoring this task, after undergoing pertinent training.

- o Maps will be in A2 for the experimentation. And they are actually tested for visual acuity to be included in Prevent due to the cognitive battery, so if they're in prevent, they have no hindering sight problems

- o The study was piloted with 2 people and the setup was adjusted accordingly.

Methods and analysis: data management - Added:

- o description of how dialogues will be transcribed and annotated, including tool used (ELAN) and data export formats.

Added section on PPI Analysis

- Added:

- o Alignment explained referring to accommodation theory and Pickering's paper
- o Dialogue features to be extracted with ELAN

- o more details on the statistical analysis to be performed in order to draw relationships between Prevent collected factors and PREVENT-ED collected factors (regression modelling and Pearson correlations)

- o Article included between predict and participant'
- o This is a longitudinal study; hence we will eventually know which people develop dementia and when and we can put this in relationship with our data

Ethics and dissemination

- ETHICS: COGNITO assessment is carried out by the main Prevent team and these data is their responsibility. They do have a clause whereby they inform participants and GPs if any clinically

relevant information arises from the research process, however this is not our competency. Our task is not validated, and no clinically relevant information will be concluded until it is further explored. This is mentioned in the ethics and dissemination section of our paper, where “all participants will be informed that this is an exploratory study and not a diagnostic test”.

Conclusions

- Reworded: consistency across data is expected, not already proved
- Known dates of the study are included in the front page. Ethics dates are included in the ethics and dissemination section of abstract and main body.

REVIEWER 2

Abstract

- Typo corrected (PREVENT)

Introduction

- Provided example for narrative speech monologues
- Clarified that the elicitive task is the map task (which also engages in spatial navigation abilities)

Methods and analysis: objective

- Elaborated rationale for cognition (interrater reliability, languages, etc.)

Methods and analysis: participants: sample size and power calculation

- Information added on which are some of the selected features and why.

Methods and analysis: experimental procedure and design

- Information added on why the task has been chosen, why it is considered more naturalistic in terms of dialogue structure than a structured conversation (which would perhaps be more naturalistic in dialogue content, but not in terms of dialogue interaction mechanisms)

Analysis

- Biomarkers and neuroimaging elaborated, as well as regression and correlation.
- Added: speech AND LANGUAGE features

Conclusions

- A few words added on the methods to quantify dialogue features (ELAN, GeMAPS, machine learning, statistical analyses).